# SepNorm: Generalization of Lion and Normalised Gradient Methods

## Abstract

In this paper, we investigate the novel optimizer Lion (Evolved Sign Momentum), which demonstrates superior performance compared to the well-established Adam in a wide range of tasks. Lion is a combination of Sign Gradient Descent (SignGD) and momentum, utilizing a fixed step size and adjusting the gradient direction via a sign operation. Despite its promising results, Lion currently lacks comprehensive theoretical justification. We also discuss Normalized Gradient Descent methods, characterized by a fixed step size, which predate Lion. We show that both Lion and NormGD have notable disadvantages, and to address these issues, we propose a new method SepNorm, which normalizes gradients across different parameter groups. SepNorm generalizes both Lion and NormGD, offering a more adaptable and stable optimization approach. Our theoretical analysis on quadratic functions reveals mechanisms of convergence behind the methods and allows us to formulate implicit bias criteria for them. Additionally, we introduce OrtSepNorm, an extension of SepNorm that makes update direction orthogonal to the weights, and we demonstrate that OrtSepNorm converges to a fixed weight norm, thereby making the training process more stable. Empirical evaluations reveal that SepNorm and OrtSepNorm outperform both Lion and Adam in a range of computer vision (CV) and natural language processing (NLP) tasks.

## 1 Introduction

The overparameterization property of modern neural networks is a key factor contributing to their high performance across a wide range of tasks. Numerous studies have demonstrated the impact of overparameterization property on the training process and loss landscape. However, it also significantly slows down training, making it computationally expensive. Therefore, the development of new optimization schemes that leverage the advantages of overparameterization while speeding up training is an important and actual problem.

Many optimization methods have been introduced specifically for neural network training. One such class of methods involves fixed step size schemes. The first proposed method was Norm (S)GD (Mandic, 2004), the key idea of which was the normalization of the gradient to the unit sphere. Another fixed step size method was Sign (S)GD (Moulay et al., 2019), which applies an element-wise sign operation to the gradient. Additionally, generalized approaches like the Block Normalized Gradient method (Yu et al., 2017) have been proposed, although they have not been deeply studied.

The authors of the paper (Chen et al., 2024) proposed a novel optimizer, Lion (Evolved Sign Momentum), discovered through program search in the symbolic representation of optimizer space. Lion combines various modifications of traditional methods with Sign (S)GD scheme: exponential smoothing with momentum, decoupled weight decay, and sign operation. Similar to Sign (S)GD, this method has a fixed step size since the norm of the output of the sign function equals the number of nonzero elements. The authors demonstrate that Lion outperforms widely used method AdamW across a various range of tasks, including language modeling and image generation with diffusion models.

However, due to the novelty of Lion, there is a lack of theoretical justification for its performance. In our paper, we investigate the mechanisms behind its effectiveness. We analyze the behavior of the method on quadratic functions and demonstrate that the sign operation establishes the lower bound on the possible loss that is dependent on the learning rate and the sharpness of the minima.

Furthermore, we introduce a new method called SepNorm, which renormalizes each group of parameters separately, similar to Block Normalized GD. This is a more general approach because Lion can be considered as SepNorm where each weight forms a separate group. For SepNorm, we also demonstrate inductive bias properties for quadratic functions. Additionally, we show that SepNorm overcomes a disadvantage of Lion, which we term "Momentum Tracing".

We also propose the OrtSepNorm method, a modification of SepNorm that projects the momentum onto a direction orthogonal to weight decay. For scale-invariant networks, we show that OrtSepNorm allows the model to first converge to a fixed weight norm, the value of which can be computed a priori, and then find better global minimum of the training loss using its implicit bias as an external criterion.

We conduct a range of experiments across different architectures and tasks, demonstrating better or comparable performance of the proposed methods over Lion and AdamW (Loshchilov, 2017). We also perform experiments in a Grokking setup (Liu et al., 2022b), confirming our theoretical hypotheses about the positive impact of equalizing convergence speed across different groups of parameters.

Thus, the main contributions of this paper are summarized as follows:

- We identify the existing drawbacks of Lion and Norm (S)GD methods and provide experimental evidence of their negative effects on the training process.
- We introduce SepNorm, a new optimization method that normalizes each group of parameters separately, providing a more general approach than Lion.
- We investigate the behavior of Lion and SepNorm on quadratic functions, demonstrating their inductive bias properties.
- We develop OrtSepNorm, a modification of SepNorm that projects momentum orthogonally to weight decay, enhancing convergence stability for scale-invariant networks.
- We conduct extensive experiments across different architectures and tasks, showing that SepNorm achieves better or comparable performance to state-of-the-art optimizers like AdamW and Lion.

## 2 RELATED WORKS

Overparameterization is a property of modern neural networks where the number of parameters is significantly greater than the number of training samples. Many studies explore the benefits of overparameterization (Belkin et al., 2019; Liu et al., 2022a; Schaeffer et al., 2023; Neyshabur et al., 2018). Another property of modern models is scale-invariance. Scale-invariance means that all or part of the parameters are invariant to multiplication by a positive constant. This property arises due to the use of normalization layers in combination with ReLU activations and has been extensively studied (Cho & Lee, 2017; Van Laarhoven, 2017; Zhang et al., 2018; Li & Arora, 2019; Li et al., 2020). In (Kodryan et al., 2022), the authors investigate the behavior of the training process of scale-invariant models with fixed weight norms and discover a strong connection between learning rate and generalization. Enhancing generalization ability is one of the core problems in deep learning. Many works try to use the sharpness of minima on the training loss as a measure of generalization (Keskar et al., 2016; Andriushchenko et al., 2023). Methods based on implicit sharpness minimization demonstrate notable improvements in performance on various tasks (Foret et al., 2020; Kwon et al., 2021; Zhuang et al., 2022).

Various optimization methods have been proposed to improve the training of neural networks. The most popular is the Adam method (Kingma, 2014), which adds to traditional SGD (Nesterov, 1983) a momentum mechanism and element-wise gradient adjustment with running statistics of gradient squares. This works as a preconditioning, allowing the model to adapt the step size independently for each component and descend down the slopes of the loss function (Cohen et al., 2022). Its modification, AdamW (Loshchilov, 2017), uses decoupled weight decay, which provides classic theoretical regularization on the weight norm. Besides, element-wise adjustments, many methods use other techniques. The Norm (S)GD method normalizes the gradient to the unit sphere, which leads to a fixed step size. Many properties of this method have been studied in (Murray et al., 2019; Mandic, 2004; Zhao et al., 2021). Another method with a fixed step size is Sign GD (Moulay et al.,

2019; Li et al., 2023; Safaryan & Richtárik, 2021). This method applies the sign operation to the update vector, which can be considered as Adam with specific relationship between moving averages between first and second order statistics. In paper (Balles et al., 2020), the authors provide deep theoretical research of Sign GD methods in terms of separable smoothness. They demonstrate that the method is sensitive to the diagonal concentration property of hessian and provide experimental evidence on quadratic functions.

The more general method Block Normalized GD, which was introduced in (Yu et al., 2017), uses block-wise gradient normalization, adding flexibility in choosing blocks for gradient adjustment.

The novel method Lion (Chen et al., 2024) was developed using program search in the symbolic representation of the optimizer space. This method can be considered as Sign GD with momentum, gradient enhancement (which can be seen as Nesterov Momentum), and decoupled weight decay. The authors demonstrate its superiority in comparison to AdamW on a wide range of tasks.

In paper (Chen et al., 2023), the authors analyze the continuous variant of Lion and introduce the family of Lion-$\mathcal{K}$ methods. Since the sign function is not differentiable, the authors use the concept of a subgradient and prove that optimization by the Lion method corresponds to conditional optimization under constraint on the weight norm:

$$\min_w L(w) \quad s.t. \ ||w||_\infty \leq \frac{1}{\lambda}, \tag{1}$$

where $\lambda$ is the weight decay hyperparameter.

However, this often holds also for traditional optimization approaches, thus it does not provide insights into the superior performance of Lion.

Many works try to provide theoretical justifications for the existing methods in application to neural networks. In paper Barrett & Dherin (2020), it was demonstrated that gradient descent implements an implicit regularization of gradient norm, which is a good sharpness measure. This also applies to stochastic methods and even to methods with adaptive steps such as Adam Smith et al. (2021), Cattaneo et al. (2023). Another interesting property strictly connected with implicit bias is the Edge of Stability Cohen et al. (2021), Cohen et al. (2022). This phenomenon relates to the behavior of the training process near the zero-loss manifold in which the model exhibits a drift according to implicit regularization flow. Research of this phenomenon for Norm (S)GD on quadratic functions demonstrates that the method has an implicit bias which regularizes the spectral norm of the Hessian Murray et al. (2019), another widely used sharpness measure.

## 3 THEORY BEHIND LION

Lion (Chen et al., 2024) was developed via program search in symbolic representation of optimizers and can be represented as following way:

**Definition 3.1.** *The update rule of Lion:*

$$\begin{cases} m_{t+1} = \beta_2 m_t + (1 - \beta_2)\nabla L(w_t) \\ w_{t+1} = (1 - \eta\lambda)w_t - \eta \operatorname{sign}(\beta_1 m_t + (1 - \beta_1)\nabla L(w_t)) \end{cases} \tag{2}$$

*where $\eta$ is learning rate, $\lambda$ - weight decay coefficient.*

Lion uses the sign operation applied to momentum adjusted gradient. Since the norm of the sign operation equals the number of nonzero elements, this often results in a fixed step size equal to $\eta\sqrt{d}$, where $d$ is the dimension of the network. The sign operation affects the update direction. Therefore, without weight decay and learning rate annealing, Lion can be considered to optimize the function on a predefined rectangular grid, significantly differing from traditional approaches.

For fixed step size methods, a more natural approach is to normalize the gradient or momentum update to its norm. This method, known as Norm (S)GD, was introduced in a paper prior to Lion and Sign (S)GD. Without the limitation of optimization on rectangular grid, this method should be more flexible in terms of update directions. However, Lion has an important advantage over Norm (S)GD. Due to the vanishing gradient problem, some weights may receive small gradient components, especially those corresponding to first layers. After the sign operation, Lion equalizes updates based

only on their sign. In Norm (S)GD, after update normalization, these small components remain small, potentially undertraining first layers compared to deeper ones with larger gradient values. Traditional methods like SGD or AdamW may also exhibit undertraining of certain layers. In the paper (Liu et al., 2022b), the authors demonstrate that if the first layers are trained slower than the latter ones, it is more likely to observe grokking phenomenon, i.e. the situation when the training loss is almost at global minimum while the validation loss remains high due to memorization of training data.

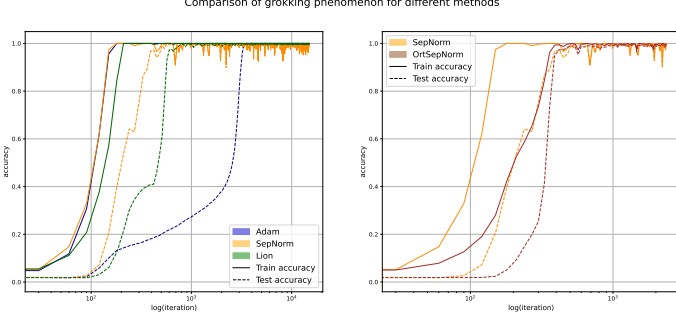

Figure 1: Comparison of Adam, Lion, SepNorm and OrtSepNorm methods on grokking setup. Grokking phenomenon is characterized by the width of the gap between points of $100\%$ accuracy on training and validation. SepNorm has the smallest gap in comparison to Adam and Lion, which indicates that the method equalizes the training speed between different layers and prevents appearance of relatively undertrained parameter groups. OrtSepNorm makes the gap even smaller by explicitly encouraging the convergence to the weights of fixed norm in addition to equalization of the training speed for all layers.

Experiments on the grokking setup confirmed our hypothesis. Figure 1(a) shows a comparison between Adam and Lion. The width of the gap between the points of $100\%$ accuracy on training and validation identifies undertraining of the first layers relative to the deeper ones. Equalization of speed in Lion allows the first layers to train faster; therefore, we observe a decrease in the gap width.

However, using the sign operation in Lion also introduces a drawback. Most modern neural networks incorporate normalization layers, such as Batch Normalization (Ioffe & Szegedy, 2015) and Layer Normalization (Ba et al., 2016), followed by ReLU or similar activations. When the preactivation feature map components are negative, ReLU outputs zero, resulting in no backpropagation signal through those components. Consequently, the corresponding components of the bias term in the normalization layer receive zero gradient.

After the gradient becomes zero, these components may continue updating in the direction of momentum for an extended period until the gradient becomes non-zero again or the momentum value diminishes to numerical precision limits. If the momentum components associated with the bias term of the normalization layers are negative, a positive feedback loop can occur. In this scenario, the bias components shift further into negative values, reducing the preactivation values even more, which in turn maintains zero gradients.

When the momentum values have already become very small and a gradient signal suddenly reappears, the weight values might have moved into suboptimal regions, causing abrupt changes that destabilize the training process. We term this effect "momentum tracing". With both convolutional and linear layers multiple parameters lose the gradient signal when the corresponding BatchNorm activation outputs zero. The experimental evidence of the effect is illustrated in 2.

In contrast to ReLU, LeakyReLU activation preserve backpropagation signal even for negative values of preactivation components, which eliminates the effect. Experiments with LeakyReLU activaions presented in Appendix C.

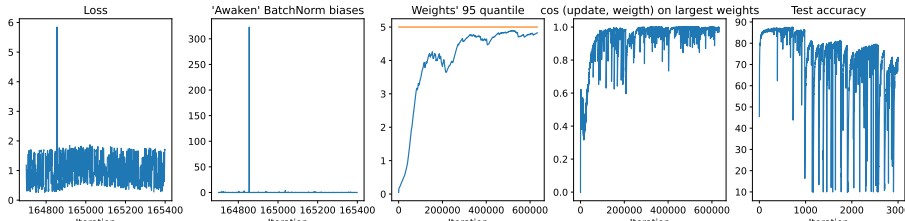

Figure 2: A display of Lion divergence with ResNet18 on CIFAR-10 . (Center left:) A number of BN biases with a near-zero momentum that receive a non-zero gradient on the current iteration. (Center:) Blue — a 95 quantile of networks' parameters values. Under Lion-weight decay dynamics weights converge to $\pm 5 = \pm 1/\lambda$ when their gradients are identical to zero. (Center right:) Cosine similarity between 5% largest weights and their updates converges to 1 as the weights converge to $1/\lambda$.

Moreover, using normalization layers with ReLU activation makes groups of weights before the activation scale-invariant, meaning that multiplication by a positive constant doesn't change the output. From this, two properties can be inferred:

$$\begin{cases} \langle \nabla L(w), w \rangle = 0, \forall w \\ \nabla L(\alpha w) = \frac{1}{\alpha} \nabla L(w) \end{cases} \tag{3}$$

The first property means that the gradient direction of the scale-invariant weights is orthogonal to the weight decay direction. It also implies that the gradient step always increases the weight norm. The second property defines the effective gradient as the gradient of the function with normalized weights: $\nabla L(\frac{w}{||w||}) = ||w|| \nabla L(w)$. As was shown in Kodryan et al. (2022) for scale-invariant networks, the norm of the effective gradient can be treated as a good approximation to sharpness of train loss on a unit sphere and is positively correlated with validation error.

Consider method Norm (S)GD. We can show interesting property:

**Theorem 3.1.** *For scale-invariant networks method Norm (S)GD without momentum converges to weight norm equal to:* $||w|| = \frac{\eta}{\sqrt{2\eta\lambda - \eta^2\lambda^2}}$

The sign operation in Lion changes the update direction, therefore even without momentum, this property will not hold. However, in practice, both methods Norm (S)GD and Lion, even with the momentum mechanism, demonstrate convergence of the weight norm although at a much slower rate.

This property is useful because theoretical guarantees for convergence to a fixed weight norm make the training process more stable. In Lobacheva et al. (2021), the authors observe periodic fluctuations in scale-invariant models with a fixed learning rate. They show that instabilities in the model can arise due to periodic explosion of weight norms. Additionally, in Kodryan et al. (2022), the authors analyze the behavior of scale-invariant models during training on a fixed sphere. In this setup, they demonstrate that the correlation between generalization and sharpness is much stronger in a fixed weight norm setup. Therefore, regularization of the sharpness after the convergence of the weight norm allows achieving minima with better generalization. Hence, it is important to explore the sharpness regularization properties of methods with fixed step sizes.

## 4 THEORY BEHIND SEPNORM

Both of the methods Lion and Norm (S)GD have different advantages and disadvantages: sign operation in Lion allows model to perform equally large update steps, even for components with small gradient norm, at the same time deflecting the update out of gradient direction, which adds additional noise to training process. In this section, we propose SepNorm, a new approach based on Block Normalized GD, which can be considered as a general version of both methods:

**Definition 4.1.** *Update of SepNorm optimization method:*

$$w_t = (1 - \eta\lambda)w_{t-1} - \eta T_s^{-1}u_t, \tag{4}$$

*where*

$$T_s = diag(\frac{||u_t^{l_1}||}{\sqrt{|l_1|}}, \frac{||u_t^{l_1}||}{\sqrt{|l_1|}}, ..., \frac{||u_t^{l_n}||}{\sqrt{|l_n|}})$$

*where $l_i$ — parameter group, $|l_i|$ — size of the group, $n$ - number of groups, $u_t$ — update vector, after momentum mechanism applying.*

In this method, the gradient is normalized separately among various groups of parameters $l_i$. Hence, Lion can be considered as SepNorm where all groups consist of a single parameter, and Norm (S)GD - as SepNorm with only one group consisting of all parameters. SepNorm uses the same weight decay and momentum mechanisms as Lion. Also, for each group of parameters their gradients are renormalized to the square root of the dimension of the corresponding group $\sqrt{|l_i|}$. This is used to align the sizes of the steps within the groups relative to the entire space, which holds for Lion. Norm of Lion's step equals to square root of non-zero gradient components, therefore renormalization with square root of group dimension equalizes Lion and SepNorm step sizes.

If some group has smaller gradients than another, for example due to a vanishing gradient, after the normalization, each group will have equal impact on the whole update of parameters. However inside each group the update direction will match gradient direction for the group.

As a choice of parameter groups, the most natural approach is to use the weights of a single layer as a separate group, which we will use in all experiments with a method. This choice allows us to equalize convergence speed among layers, which prevents underfitting of some part of the network. However, for some models it may be beneficial to join several layers into blocks and handle it as a separate group.

For the proposed method, the overall direction will not differ as much as it does in Lion, which means that SepNorm is a less noisy method than Lion. The noisiness of Lion is implicitly confirmed by the fact that Lion prefers larger batch sizes. The value of the batch size controls the level of noise, and the noise from small batches in addition to noise from the sign operation leads to suboptimal behavior for Lion. SepNorm is devoid of such problems since the noise from block normalization is significantly less than in Lion. Thus, SepNorm is a more flexible method in terms of hyperparameter selection and can show better performance with small batch sizes.

For method SepNorm without momentum, holds the similar property as for Norm (S)GD. However, the proof of this is based on the following lemma (proof in Appendix A.):

**Lemma 4.1.** *For each scale-invariant group $l_i$ holds scale-invariance property of orthogonality:*

$$\langle \nabla_{l_i} L(w), w_{l_i} \rangle = 0$$

Then, the validity of the following theorem is obvious (proof in Appendix A.):

**Theorem 4.2.** *Consider method SepNorm without momentum on scale-invariant networks with normalization among scale-invariant groups. Then, weight norm of each group $l_i$ converges to:*

$$||w_{l_i}|| = \frac{\eta\sqrt{|l_i|}}{\sqrt{2\eta\lambda - \eta^2\lambda^2}}$$

In the majority of architectures, for weights in a single layer holds the scale-invariance property, which in turn justifies our choice to use layers as separate groups for SepNorm.

We also introduce OrtSepNorm, which is a modification of SepNorm for explicitly controlling the weight norm in scale-invariant networks

**Definition 4.2.** *Update of OrtSepNorm optimization method:*

$$\begin{cases} g = \beta_1 m_t + (1 - \beta_1)\nabla L(w_t) \\ g = g - \cos(w_t, g)\frac{||g||}{||w_t||}w_t \\ w_{t+1} = (1 - \eta\lambda)w_t - \eta T_s^{-1}g \\ m_{t+1} = \beta_2 m_t + (1 - \beta_2)\nabla L(w_t) \end{cases} \tag{5}$$

The modified method has the property that its updates for each group of parameters are always orthogonal to the weight direction in scale-invariant networks. Therefore, the following theorem holds (proof in Appendix A.):

**Theorem 4.3.** *Method OrtSepNorm converges to weight norm equal to:*

$$||w|| = \frac{\eta\sqrt{d}}{\sqrt{2\eta\lambda - \eta^2\lambda^2}}$$

Thus, OrtSepNorm has explicit theoretical guarantees for convergence to a fixed weight norm. In experiments with grokking, OrtSepNorm showed the smallest gap (Figure 1(b)) among other methods due to faster weight norm convergence. And after the weight norm has stabilised, the implicit regularization of sharpness (see next section) of fixed step size methods encourages convergence to a global minimum with better generalization.

## 5 GEOMETRICAL INTERPRETATION OF METHODS ON QUADRATIC FUNCTIONS

In this section, we will analyze the behaviour of Sign GD and SepNorm methods without momentum on quadratic functions. Let $L(w) = \frac{1}{2}w^T A w$ be a quadratic function with some positive definite matrix $A$. Let $\lambda_{\max}(A) = \lambda_1 \geq \lambda_2 \geq ...\lambda_d$ - eigenvalues of $A$, $v_1, ..., v_d$ - corresponding eigenvectors. After the convergence the method starts oscillating between two points that can be regarded as an attractor for the given optimizer. Let $L^{\max} = \max(L(w_t), L(w_{t+1}))$, $t \gg 1$. Of course the particular attractor will depend on the choice of initial point $w_0$ as well as the stochasticity of the trajectory $\tau$ caused by the use of stochastic gradients. Denote $\mathbb{E}L^{\max}$ the expected value of $L^{\max}$ w.r.t. to all possible $\tau$ and $w_0$.

The important results from Arora et al. (2022) is that Norm GD converges to an attractor and loss starts oscillating between points $\frac{1}{2}C^2\lambda_1\eta^2$ and $\frac{1}{2}(C-1)^2\lambda_1\eta^2$, where $0 < C < 1$. This allows us to write a lower bound for $L^{\max}$ for Norm GD (proof in Appendix B.):

**Lemma 5.1.** *For Norm GD without momentum following lower bound holds:*

$$\lambda_{\max}(A) \leq \frac{8}{\eta^2}\mathbb{E}L^{\max} \tag{6}$$

To sum up, Norm GD oscillates along the eigenvector that corresponds to the largest eigenvalue, i.e. sharpest direction.

To derive a lower bound for Sign GD, first we have to prove the following lemma (proof in Appendix B.):

**Lemma 5.2.** *Consider the function $L(w) = \frac{1}{2}w^T A w$, with a positive definite diagonal matrix $A$. Then, for Sign GD for any choice of $w_0$ and optimization trajectory $\tau$ it holds:*

$$Tr(A) \leq \frac{8}{\eta^2}L^{\max}. \tag{7}$$

*The bound is tight and corresponds to $w_t = -w_{t+1} = \frac{\eta}{2}e$, where $e$ is a vector consisting of $+1$ and $-1$.*

This lemma allows us to formulate the main result for the behavior of Sign GD on quadratic functions (proof in Appendix B.):

**Theorem 5.3.** *Consider the function $L(w) = \frac{1}{2}w^T A w$, with a positive definite matrix $A$. Then, for Sign GD it holds:*

$$Tr(A) \leq \frac{8}{\eta^2}\mathbb{E}L^{\max} \tag{8}$$

This theorem sets a lower bound on the average loss value to which the method converges. This bound depends on the trace of the Hessian (a spectral sharpness measure). In other words, in sharp regions, the method cannot achieve low loss values. Therefore, the learning rate and local sharpness control the loss value to which the method may converge.

SepNorm combines the properties of both Norm GD and Sign GD. The update rule can be considered independently for each block, which means that weights inside blocks will be optimized as Norm GD. Therefore, inside each block, the method will converge to the eigenvector corresponding to the largest eigenvalue of the block. Then we may apply result from Sign GD treating sharpest direction within each block as new variable in Sign GD (proof in Appendix B.):

**Theorem 5.4.** *Consider the function* $L(w) = \frac{1}{2}w^T A w$, *with a positive definite matrix* $A$. *Let* $l_i, i = \{1, ..., n\}$ *be a set of blocks of normalization in SepNorm. Let* $A_i$ *be the block of the matrix* $A$ *corresponding to* $l_i$. *Then, for SepNorm it holds:*

$$\sum_{i=1}^{n} \lambda_{\max}(A_i)|l_i| \leq \frac{8}{\eta^2}\mathbb{E}L^{\max} \tag{9}$$

Thus, we can conclude that all considered methods possess the following feature: they can converge to low loss values only in the regions with low sharpness, thus establishing the inductive bias towards wide minima.

# 6 Experiments

In this section, we compare Lion, AdamW, SepNorm, and OrtSepNorm on CV and NLP tasks. The selection of these optimizers is based on AdamW's status as the most popular method for training large neural networks, and Lion's superior performance on a wide range of tasks.

All the experiments described below were done in mixed precision. Results are averaged over three runs for NLP and CV fine-tuning tasks, and over five runs for Imagenet full training. All the hyperparameters are specified in the Appendix D.

## 6.1 Image Classification

The authors of the paper on Lion evaluated the performance of Lion and AdamW optimizers on transformer architectures, varying the batch size. They concluded that the optimal batch size for AdamW is 256, whereas for Lion, it is 4096. Given that our method offers less noisy updates compared to Lion, we conducted our comparison using a batch size of 256. For training on computer vision tasks, we used Vision Transformers (ViT) pre-trained on ImageNet-21k to save time. The neural network training spanned 100,000 steps, incorporating a 4,000-step warmup phase, with the learning rate following a cosine annealing schedule. More details about the hyperparameters used for training can be found in Appendix D. As shown in Table 1, training with SepNorm achieved better accuracy than other tested methods and, as expected, the Lion optimizer performed slightly better than AdamW.

Furthermore, we trained transformers on Imagenet and V2 tasks using the augmentation procedure presented by Touvron et al. (2022). Training was conducted with a batch size of 64, and the results are presented in Table 1 with the prefix "DeiT".

We also trained ResNet-50 from scratch. As discussed earlier in the theoretical section, ResNet architectures are poorly compatible with the Lion optimizer when training with a small batch size due to the "Momentum Tracing" effect. Consequently, in this experiment, we provide a comparison only with AdamW. As shown in Table 2, SepNorm achieves superior accuracy on the test set.

To explore how the batch size affects the performance of the methods, we conducted ResNet-50 experiments with varying batch sizes. We employed linear scaling for the learning rate and kept the weight decay constant. As shown in Table 2, AdamW peaks at a batch size of 256, while SepNorm achieves its strongest performance at a larger batch size of 512. Regrettably, we did not have sufficient resources to test larger batch sizes.

## 6.2 Language Modeling and Fine-tuning

We fine-tuned 11B T5 models on the GLUE benchmark. Each model underwent fine-tuning for 50,000 steps, using a batch size of 128 and a constant learning rate. Due to limited computing resources, we used 10 times fewer steps than the authors of Lion, which far exceeded the validation

Table 1: Pre-train on ImageNet-21K then fine-tune on ImageNet

|  | Optimizer | Imagenet | ReaL | V2 |
|---|---|---|---|---|
| ViT-B/16$_{384}$ | AdamW | 83.75 | 88.01 | 73.33 |
|  | Lion | 83.90 | 88.27 | 73.41 |
|  | Sepnorm | **84.18** | **88.56** | **73.66** |
| ViT-S/16$_{384}$ | AdamW | 78.01 | 83.89 | 66.24 |
|  | Lion | 78.13 | 83.00 | 66.39 |
|  | Sepnorm | **78.35** | **84.31** | **66.72** |
| Mixer-B/16$_{384}$ | AdamW | 73.98 | 80.11 | 59.87 |
|  | Lion | 74.04 | 80.15 | 59.98 |
|  | Sepnorm | **74.19** | **80.46** | **60.21** |
| Deit ViT-S/16$_{384}$ | AdamW | 83.56 | – | 73.21 |
|  | Lion | 83.44 | – | 73.19 |
|  | Sepnorm | **83.67** | – | **73.35** |
| Deit ViT-B/16$_{224}$ | AdamW | 83.80 | – | 73.62 |
|  | Lion | 83.72 | – | 73.55 |
|  | Sepnorm | **83.93** | – | **73.76** |

Table 2: Validation accuracy for full training of ResNet50 on Imagenet with varying batch sizes.

| batch size | AdamW | SepNorm |
|---|---|---|
| 64 | 75.894 | **76.429** |
| 256 | 76.137$_{\pm0.072}$ | **76.460$_{\pm0.107}$** |
| 512 | 75.382 | **76.626** |

peak step for almost all tasks. As shown in Table 3, SepNorm outperforms all other optimizers on the tasks CoLA, MRPC, STS-B, QQP, MNLI-m, and RTE. For other tasks (SST-2, MNLI-mm, QNLI), both Lion and SepNorm show similar performance. We report the default metrics for CoLA, SST-2, MNLI-m, MNLI-mm, QNLI, and RTE. For MRPC and QQP, we provide both F1 and Accuracy scores. For STS-B, we report the Pearson and Spearman correlations.

Table 3: Fine-tuning performance of the T5 Base on the GLUE dev set. Results reported are the peak validation scores per task.

| Task | AdamW | Lion | SepNorm | OrtSepNorm |
|---|---|---|---|---|
| CoLA | 61.5$_{\pm1.6}$ | 60.7$_{\pm0.7}$ | **61.8$_{\pm1.1}$** | **61.3$_{\pm1.4}$** |
| SST-2 | 94.6$_{\pm0.3}$ | **94.8$_{\pm0.4}$** | 94.6$_{\pm0.3}$ | **94.9$_{\pm0.1}$** |
| MRPC | 93.9$_{\pm0.3}$ / 91.3$_{\pm0.5}$ | 93.8$_{\pm0.2}$ / 91.2$_{\pm0.3}$ | **94.2$_{\pm0.4}$/92.0$_{\pm0.7}$** | **94.2$_{\pm0.4}$/91.9$_{\pm0.6}$** |
| STS-B | 90.4$_{\pm0.1}$ / 90.3$_{\pm0.0}$ | 90.7$_{\pm0.1}$ / 90.5$_{\pm0.0}$ | **90.9$_{\pm0.4}$/90.7$_{\pm0.3}$** | **91.2$_{\pm0.1}$/90.9$_{\pm0.1}$** |
| QQP | 88.4$_{\pm0.1}$ / 91.2$_{\pm0.1}$ | 88.2$_{\pm0.2}$/91.1$_{\pm0.1}$ | **88.4$_{\pm0.2}$/91.2$_{\pm0.1}$** | **88.3$_{\pm0.2}$/91.1$_{\pm0.1}$** |
| MNLI -m | 87.0$_{\pm0.4}$ | 86.9$_{\pm0.3}$ | **87.0$_{\pm0.1}$** | **87.1$_{\pm0.1}$** |
| MNLI -mm | 87.1$_{\pm0.1}$ | 87.1$_{\pm0.1}$ | **87.2$_{\pm0.1}$** | **87.2$_{\pm0.1}$** |
| QNLI | 92.9$_{\pm0.0}$ | 93.0$_{\pm0.2}$ | **93.0$_{\pm0.1}$** | 92.9$_{\pm0.1}$ |
| RTE | 79.9$_{\pm1.0}$ | 79.5$_{\pm0.3}$ | **80.4$_{\pm0.6}$** | **80.4$_{\pm0.7}$** |

## 7 CONCLUSION

In this paper, we introduce and analyze SepNorm and OrtSepNorm, new optimization methods designed to address the shortcomings of Lion and Norm GD. Our theoretical and empirical evaluations have revealed certain drawbacks of Lion. By examining the Lion alongside another fixed step size method, Norm GD, we outline the major properties of both methods, which led us to develop the SepNorm scheme. Specifically, the combination of a sign operation with momentum in Lion led to instabilities due to a phenomenon we termed "Momentum Tracing." Furthermore, we analyze the application of our methods to scale-invariant networks and demonstrate that they can converge to a fixed weight norm, thereby stabilizing the training process. Additionally, we provide a theoretical investigation into the behavior of these methods on quadratic functions, which helps us define their implicit bias properties. Empirical evaluations across various practical setups show that the proposed SepNorm methods outperform both Adam and Lion.

**Limitations.** OrtSepNorm showed superior performance only in NLP tasks. Experiments on CV tasks with ResNet architectures demonstrate suboptimal performance for the method due to the use of a scheduler that causes the weight norm to become too low values, which in turn introduces instabilities to the training process. to the training process. Also, we provide theoretical results only for quadratic functions, which may be insufficient to generalize our theoretical findings to neural networks.

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

## A  PROOFS OF CONVERGENCE TO SPHERE

In this section, we provide proofs for the theorems about weight norm convergence on scale-invariant networks. We will use the following notation: $\eta$ is the learning rate, $\lambda$ is the weight decay coefficient.

Let's formally recall all methods.

**Definition A.1.** *The update rule of Lion:*

$$\begin{cases} m_{t+1} = \beta_2 m_t + (1-\beta_2)\nabla L(w_t) \\ w_{t+1} = (1-\eta\lambda)w_t - \eta \operatorname{sign}(\beta_1 m_t + (1-\beta_1)\nabla L(w_t)) \end{cases} \tag{10}$$

*where $\eta$ is learning rate, $\lambda$ - weight decay coefficient, $\beta_1$ and $\beta_2$ are hyperparameters for momentum..*

**Definition A.2.** *The update rule of Norm (S)GD with decoupled weight decay and momentum:*

$$\begin{cases} w_{t+1} &= (1-\eta\lambda)w_t + \eta\frac{\beta_1 m_t - (1-\beta_1)\nabla L(w_t)}{||\beta_1 m_t - (1-\beta_1)\nabla L(w_t)||} \\ m_{t+1} &= \beta_2 m_t + (1-\beta_2)\nabla L(w_t). \end{cases} \tag{11}$$

**Definition A.3.** *Update of SepNorm optimization method:*

$$w_t = (1-\eta\lambda)w_{t-1} - \eta T_s^{-1}u_t, \tag{12}$$

*where*

$$T_s = diag(\frac{||u_t^{l_1}||}{\sqrt{|l_1|}}, \frac{||u_t^{l_1}||}{\sqrt{|l_1|}}, ..., \frac{||u_t^{l_n}||}{\sqrt{|l_n|}})$$

*where $l_i$ is a parameter group, $|l_i|$ is the size of the group, $n$ is the number of groups, and $u_t$ is the update vector after applying the momentum mechanism.*

**Definition A.4.** *Update of OrtSepNorm optimization method:*

$$
\begin{cases}
g = \beta_1 m_t + (1 - \beta_1)\nabla L(w_t) \\
g = g - \cos(w_t, g)\frac{||g||}{||w_t||}w_t \\
w_{t+1} = (1 - \eta\lambda)w_t - \eta T_s^{-1}g \\
m_{t+1} = \beta_2 m_t + (1 - \beta_2)\nabla L(w_t)
\end{cases} \tag{13}
$$

The scale-invariance property imposes the following important properties on the gradient:

$$
\begin{cases}
\langle \nabla L(w), w \rangle = 0, \forall w \\
\nabla L(\alpha w) = \frac{1}{\alpha}\nabla L(w)
\end{cases} \tag{14}
$$

The first property, that gradient is orthogonal to weight decay direction, is crucial for following proofs.

Firstly, let's prove the theorem of convergence for Norm GD as the simplest method:

**Theorem 4.1.** *For scale-invariant networks, the method Norm GD without momentum converges to a weight norm equal to:* $||w|| = \frac{\eta}{\sqrt{2\eta\lambda - \eta^2\lambda^2}}$

*Proof.* Let's write the gradient update of Norm GD without momentum:

$$
w_{t+1} = (1 - \eta\lambda)w_t - \eta\frac{\nabla L(w)}{||\nabla L(w)||}
$$

Then, due to the scale-invariance property

$$
||w_{t+1}||^2 = (1 - \eta\lambda)^2||w_t||^2 + \eta^2
$$

After the stabilization of the weight norm, we can write it as:

$$
||w||^2 = (1 - \eta\lambda)^2||w||^2 + \eta^2
$$

Therefore, after simplification:

$$
||w||^2 = \frac{\eta^2}{2\eta\lambda - \eta^2\lambda^2}
$$

$\square$

The proof of the same theorem for SepNorm requires the following lemma:

**Lemma 4.2.** *For each scale-invariant group $l_i$, the scale-invariance property holds:*

$$
\langle \nabla_{l_i} L(w), w_{l_i} \rangle = 0
$$

*Proof.* If $l_i$ is a scale-invariant group of parameters, we can freeze the other weights in the neural network and consider only the dependence on these parameters. Since multiplying $l_i$ by a positive constant will not change the result, the overall function is scale-invariant, and the orthogonality property will hold for the group. $\square$

Then, the following theorem for SepNorm is valid:

**Theorem 4.3.** *Consider the method SepNorm without momentum on scale-invariant networks with normalization among scale-invariant groups. Then, the weight norm of each group $l_i$ converges to:*

$$
||w_{l_i}|| = \frac{\eta\sqrt{|l_i|}}{\sqrt{2\eta\lambda - \eta^2\lambda^2}}
$$

*Proof.* Consider the gradient update of the SepNorm method:

$$w_{t+1} = (1 - \eta\lambda)w_t - \eta T_s^{-1} u_t$$

Let's split the update rule into blocks and consider the update rule for them separately:

$$w_{t+1}^{l_i} = (1 - \eta\lambda)w_t^{l_i} - \eta \frac{u_t^{l_i}}{||u_t^{l_i}||} \sqrt{|l_i|}$$

Taking the norm from both sides and using the previous lemma gives:

$$||w_{t+1}^{l_i}||^2 = (1 - \eta\lambda)^2 ||w_t^{l_i}||^2 + \eta^2 |l_i|$$

Therefore, by analogy with the theorem for Norm GD:

$$||w_{l_i}|| = \frac{\eta\sqrt{|l_i|}}{\sqrt{2\eta\lambda - \eta^2\lambda^2}}$$

$\square$

Explicit orthogonalization in OrtSepNorm allows us to formulate a similar theorem without the scale-invariance property of models:

**Theorem 4.4.** *The method OrtSepNorm converges to a weight norm equal to:*

$$||w|| = \frac{\eta\sqrt{d}}{\sqrt{2\eta\lambda - \eta^2\lambda^2}}$$

*Proof.* The proof for the theorem is similar to SepNorm. Here, the additional term with the dot product between the update and weight vector is eliminated due to explicit projection to the orthogonal direction. $\square$

# B PROOFS FOR GEOMETRICAL INTERPRETATION OF METHODS ON QUADRATIC FUNCTIONS

In this section, we provide theoretical proofs for the proposed theorems.

The paper Arora et al. (2022) shows that for a quadratic function, the Norm GD method converges and oscillates between points with loss values $\frac{1}{2}C^2\lambda_1\eta^2$ and $\frac{1}{2}(1 - C)^2\lambda_1\eta^2$, for some $0 < C < 1$. Based on this results, we can get a lower bound for $L^{\max}$:

**Lemma 5.1.** *For Norm GD without momentum, the following lower bound holds:*

$$\lambda_{\max}(A) \leq \frac{8}{\eta^2}\mathbb{E}L^{\max} \tag{15}$$

*Proof.* If the loss oscillates between $\frac{1}{2}C^2\lambda_1\eta^2$ and $\frac{1}{2}(1 - C)^2\lambda_1\eta^2$, then it is evident that:

$$L^{\max} = \max(\frac{1}{2}(C - 1)^2\lambda_1\eta^2, \frac{1}{2}C^2\lambda_1\eta^2) \geq \frac{1}{8}\lambda_1\eta^2$$

$\square$

Now, consider the lemma that helps us to prove the main result for Sign GD.

**Lemma 5.2.** *Consider the function $L(w) = \frac{1}{2}w^T A w$, with a positive definite diagonal matrix $A$. Then, for Sign GD, for any choice of $w_0$ and optimization trajectory $\tau$, the following holds:*

$$Tr(A) \leq \frac{8}{\eta^2}L^{\max}. \tag{16}$$

*The bound is tight and corresponds to $w_t = -w_{t+1} = \frac{\eta}{2}e$, where $e$ is a vector consisting of $+1$ and $-1$.*

*Proof.* We aim to find the minimal value for $L^{\max}$. For a quadratic function with diagonal matrix $A$, it is evident that the optimal points $w$ and $w'$, between which the function oscillates, lie in region $w^i \in (-\eta, \eta)$, where $\eta$ is a learning rate. Moreover, if function oscillates between two points on a line, then the minimum of $L^{\max}$ is achieved if and only if $w = -w'$. This is because, in Sign GD, we can only move along the diagonals, and quadratic functions have symmetric properties along $y = x$ and $y = -x$.

Using this property allows us to find the optimal points:

$$\frac{1}{2} w^T A w = \frac{1}{2} (w - \eta \operatorname{sign}(Aw))^T A (w - \eta \operatorname{sign}(w))$$

Then:

$$\eta \operatorname{sign}(Aw)^T A (w - \frac{1}{2} \eta \operatorname{sign}(Aw)) = 0$$

This equation can be holds either if $\eta \operatorname{sign}(Aw)^T A$ is orthogonal to $(w - \frac{1}{2} \eta \operatorname{sign}(Aw))$, or if $w = \frac{1}{2} \eta \operatorname{sign}(Aw)$. However, the first case may occur only if, after the optimization step, we shift from one quadrant to another non-symmetric quadrant. If $w' = w - \frac{1}{2} \eta \operatorname{sign}(Aw)$ and $\operatorname{sign}(Aw)_i = \operatorname{sign}(Aw')_i$, then $\operatorname{sign}(Aw)_i * (Aw')_i \geq 0$ and dot product is not 0. Thus, $\operatorname{sign}(Aw)_i \neq \operatorname{sign}(Aw')_i$ and the next step will not return us to the previous point; therefore, the method has not converged yet. Consider the second case, and denote $e = \operatorname{sign}(Aw)$:

$$w = \frac{1}{2} \eta e$$

Now, compute the value of the function:

$$L = \frac{1}{8} \eta^2 e^T A e$$

But the quadratic form of a diagonal matrix $A$ evaluated at vectors of $\{+1, -1\}$ is equal to $\operatorname{Tr}(A)$. Taking into account that we considered the minimal possible value of $L^{\max}$, we can obtain the desired bound:

$$L^{\max} \geq \frac{1}{8} \eta^2 \operatorname{Tr}(A)$$

$\square$

Therefore, we state the main theorem:

**Theorem 5.3.** *Consider the function $L(w) = \frac{1}{2} w^T A w$, with a positive definite matrix $A$. Then, for Sign GD, the following holds:*

$$Tr(A) \leq \frac{8}{\eta^2} \mathbb{E} L^{\max} \tag{17}$$

*Proof.* Without a loss of generality let $\eta = 1$. The previous lemma allows us to get this lower bound for any positive definite diagonal matrix $A$. Now consider an arbitrary positive matrix $A$ with the same eigenvalues. This matrix can be seen as an image of the diagonal matrix under rotation. Sign operation divides $\mathbb{R}^d$ onto $2^d$ sectors $\{S_e\}$ with $e = (\pm 1, \ldots, \pm 1)$, where $S_e$ contains points $w$ whose update vector is $\operatorname{sign}(Aw) = e$. Analogically to previous proof, suppose a trajectory $\tau$ stabilized on two points $a$ and $b$, that correspond to update vectors $\pm e$ and sector $S_e$ correspondigly. Following the previous lemma we have

$$L^{\max} \geq \frac{1}{8} e^T A e.$$

Observe that

$$\frac{1}{2^d} \sum_{e = (\pm 1, \ldots, \pm 1)} e^T A e = \operatorname{Tr}(A)$$

for any matrix $A$. The above expression gives us a prove in a case when events $\{a \in S_e\}$ are equiprobable for every $e$.

For a diagonal matrix they are indeed equiprobable. After the rotation, a solid angle of sectors corresponding to update vectors $e$ with smaller values $e^T A e$ will decrease. Indeed, let's consider a sector $S_e$ corresponding to an update vector $e = (\pm 1, \ldots, \pm 1)$ under quadratic form $A$:

$$S_e = \{w \in \mathbb{R}^d \mid \text{sign}(Aw) = e\} = A^{-1}(\{x \mid \text{sign}(w) = e\}).$$

A sector $S_e$ is therefore described by $n$ vectors of a form $A^{-1}((0, \ldots, 1, \ldots, 0)) = v_i$. Denote $m_e = A^{-1}(e)/d$ the center of mass of $\{v_i\}$. It is straightforward to show that for a vector $e$ having a smaller $e^T A e$ value the corresponding $m_e$ has a larger norm. Observe that $m_e \sqrt{d}$ belongs to the image of the standard sphere $\mathbb{S}^{d-1}$ under $A^{-1}$. Since a volume of $S_e \cap A^{-1}(\mathbb{D}^d)$ does not depend on $e$, a solid angle of $S_e$ is inclined to be smaller for $m_e$ with a larger norm.

The expectation $\mathbb{E}L^{\max}$ is computed by averaging over all possible trajectories. Sectors $S_e$ with larger norms are not stable attractors; small perturbations can shift the point to flatter ones where it may begin to oscillate symmetrically. Considering that the probability distribution is characterized by the norm of $m_e$, it follows that among all trajectories, the probability of convergence to attractors corresponding to steeper update directions will be larger than for attractors corresponding to flatter ones. Therefore, we have that $\frac{8}{\eta^2}\mathbb{E}L^{\max}$ is larger for the rotated matrix $A$ than it is for the diagonal matrix.

$\square$

Considering the previous results, we can also prove the validity of the following lower bound for SepNorm:

**Theorem 5.4.** *Consider the function $L(w) = \frac{1}{2}w^T A w$, with a positive definite matrix $A$. Let $l_i, i = \{1, \ldots, n\}$ be a set of normalization blocks in SepNorm. Let $A_i$ be the block of the matrix $A$ corresponding to $l_i$. Then, for SepNorm, the following holds:*

$$\sum_{i=1}^{n} \lambda_{\max}(A_i)|l_i| \leq \frac{8}{\eta^2}\mathbb{E}L^{\max} \tag{18}$$

*Proof.* Firstly, let $A$ be a blockwise diagonal matrix. This property allows us to rewrite our function in the following form:

$$L(w) = \frac{1}{2}w^T A w = \sum_{i=0}^{n} \frac{1}{2}w_i^T A_i w_i = \sum_{i=0}^{n} L_i(w_i)$$

At each step, the gradients for all $w_i$ are independent. Therefore, if we consider the optimization of the function $L_i(w_i)$, we get the Norm GD method with a step size equal to $\eta' = \eta\sqrt{|l_i|}$. Due to the previous results, Norm GD will converge to the eigenvector corresponding to the largest eigenvalue of $A_i$, and the loss will oscillate between $\frac{1}{2}C^2\lambda_{\max}(A_i)\eta'^2$ and $\frac{1}{2}(1-C)^2\lambda_{\max}(A_i)\eta'^2$. Then, we get the following lower bound:

$$\lambda_{\max}(A_i) \leq \frac{8}{\eta'^2}\mathbb{E}L_i^{\max} = \frac{8}{\eta^2|l_i|}\mathbb{E}L_i^{\max}$$

Then, the lower bound for the initial function $L(w)$:

$$\frac{8}{\eta^2}\mathbb{E}L^{\max} = \frac{8}{\eta^2}\sum_{i=0}^{n} L_i^{\max} \geq \sum_{i=0}^{n} |l_i|\lambda_{\max}(A_i)$$

After convergence in each block, the function will oscillate along the largest eigenvector, i.e., the sharpest direction. This corresponds to convergence in the whole space along some of the sharpest directions. Now, if we consider an arbitrary matrix $A$ that can be obtained by some rotation of diagonal ones with same eigenvalues, similar to the proof for Sign GD, the probability of convergence to sharp attractors will decrease. This means that the method will more likely converge to the flattest sectors, in which the minimal $L^{\max}$ function will be significantly higher than $L^{\max}$ corresponding to the blockwise diagonal matrix. Therefore, the blockwise diagonal case will have a smaller $\mathbb{E}L^{\max}$ as in Sign GD. In turn, this means that the proven lower bound will hold for an arbitrary matrix $A$. $\square$

## C    MOMENTUM TRACING

To get additional experimental evidence of "Momentum Tracing", we have conducted an experiment on ResNet56, where we replace ReLU with LeakyReLU. LeakyReLU almost never has a zero derivative, and in this setup Lion does not destabilise, but is still less performant than SepNorm (please see the attached graph).

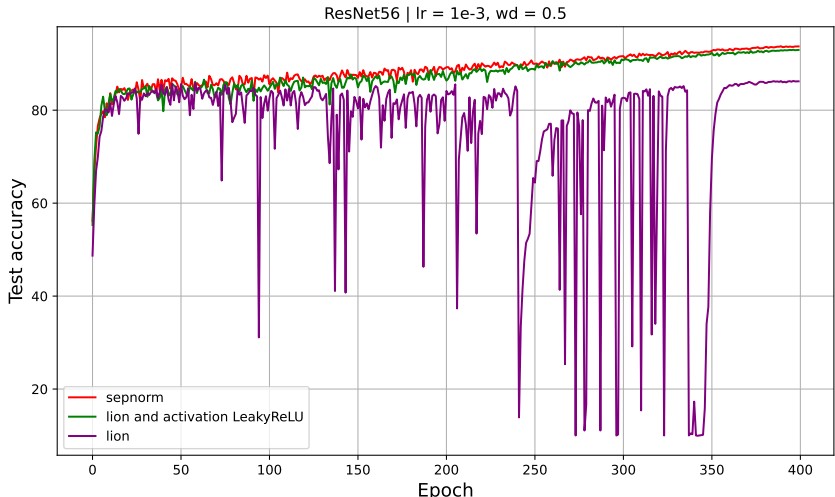

Figure 3: Implementation of LeakyReLU activation allows to significantly reduces destabilization for Lion

## D    HYPERPARAMETERS

Table 4: Hyperparameters for Grokking task.

| Model | Optimizer | $\beta_1$ | $\beta_2$ | $lr$ | $\lambda$ | steps |
|-------|-----------|-----------|-----------|------|-----------|-------|
| | AdamW | 0.9 | 0.98 | $1e-3$ | 1 | 15000 |
| Grokking | Lion | 0.9 | 0.99 | $1e-3$ | 5 | 15000 |
| | Sepnorm | 0.9 | 0.99 | $1e-3$ | 5 | 15000 |
| | OrtSepnorm | 0.9 | 0.99 | $1e-3$ | 0.1 | 15000 |

Table 5: Hyperparameters for Resnet50 with batch size 256.

| Model | Optimizer | $\beta_1$ | $\beta_2$ | $lr$ | $\lambda$ | epoch | scheduler | warmup (epoch) |
|-------|-----------|-----------|-----------|------|-----------|-------|-----------|----------------|
| | AdamW | 0.9 | 0.999 | $7.5e-4$ | 0.1 | 100 | cosine | 5 |
| Resnet50 | Lion | 0.9 | 0.99 | $7.5e-5$ | 1 | 100 | cosine | 5 |
| | Sepnorm | 0.9 | 0.99 | $7.5e-5$ | 1 | 100 | cosine | 5 |

Table 6: Hyperparameters for Vit models.

| Model | Optimizer | $\beta_1$ | $\beta_2$ | $lr$ | $\lambda$ | steps | scheduler | warmup (step) | batch size |
|---|---|---|---|---|---|---|---|---|---|
| | AdamW | 0.9 | 0.999 | $6.25e-5$ | 0.1 | 100000 | cosine | 4000 | 256 |
| ViT-S/16$_{384}$ | Lion | 0.9 | 0.99 | $6.25e-6$ | 0.3 | 100000 | cosine | 4000 | 256 |
| | Sepnorm | 0.9 | 0.99 | $6.25e-6$ | 0.3 | 100000 | cosine | 4000 | 256 |
| | AdamW | 0.9 | 0.999 | $6.25e-5$ | 0.1 | 100000 | cosine | 4000 | 256 |
| ViT-B/16$_{384}$ | Lion | 0.9 | 0.99 | $6.25e-6$ | 0.3 | 100000 | cosine | 4000 | 256 |
| | Sepnorm | 0.9 | 0.99 | $6.25e-6$ | 0.3 | 100000 | cosine | 4000 | 256 |
| | AdamW | 0.9 | 0.999 | $1e-3$ | 0.3 | 100000 | cosine | 4000 | 256 |
| Mixer-B/16$_{384}$ | Lion | 0.9 | 0.99 | $3e-4$ | 3 | 100000 | cosine | 4000 | 256 |
| | Sepnorm | 0.9 | 0.99 | $3e-4$ | 3 | 100000 | cosine | 4000 | 256 |

Table 7: Hyperparameters for Deit Vit models.

| Model | Optimizer | $\beta_1$ | $\beta_2$ | $lr$ | $\lambda$ | epoch | scheduler | warmup (epoch) | batch size |
|---|---|---|---|---|---|---|---|---|---|
| | AdamW | 0.9 | 0.999 | $1e-5$ | 0.1 | 20 | cosine | 5 | 64 |
| Deit ViT-S/16$_{384}$ | Lion | 0.9 | 0.99 | $1e-6$ | 0.3 | 20 | cosine | 5 | 64 |
| | Sepnorm | 0.9 | 0.99 | $1e-6$ | 0.3 | 20 | cosine | 5 | 64 |
| | AdamW | 0.9 | 0.999 | $1e-5$ | 0.1 | 20 | cosine | 5 | 64 |
| Deit ViT-B/16$_{384}$ | Lion | 0.9 | 0.99 | $1e-6$ | 0.3 | 20 | cosine | 5 | 64 |
| | Sepnorm | 0.9 | 0.99 | $1e-6$ | 0.3 | 20 | cosine | 5 | 64 |

