# OpenReview forum: "SepNorm: Generalization of Lion and Normalised Gradient Methods"
_ICLR.cc/2025/Conference — Submitted to ICLR 2025_

### Official Review · Reviewer_k9np · 2024-10-28

**Soundness:** 3
**Presentation:** 2
**Contribution:** 2
**Rating:** 3
**Confidence:** 4

**Summary:**

The paper introduces SepNorm and OrtSepNorm, novel optimizers aiming to generalize (recent) Lion and NormGD optimizers. Lion, while effective across a variety of tasks, lacks theoretical foundations and faces limitations such as “momentum tracing,” where layers receive zero gradients under certain conditions. SepNorm extends Lion by normalizing gradients across parameter groups rather than individually, enhancing stability. OrtSepNorm further improves convergence stability by projecting the update direction orthogonal to the weight, achieving a fixed weight norm. Experiments across CV and NLP tasks demonstrate that SepNorm and OrtSepNorm outperform Lion and Adam optimizers.

**Strengths:**

1. The paper is well-written, and the motivation of the study is clear.

2.  SepNorm and OrtSepNorm generalize existing methods (Lion, NormGD) with a theoretically approach.

3. The paper presents an in-depth theoretical analysis on the convergence properties of these optimizers on quadratic functions, identifying implicit biases and stability mechanisms.

4. Experiments show SepNorm’s and OrtSepNorm’s advantages over Lion and AdamW in terms of generalization, convergence speed, and robustness to batch size.

**Weaknesses:**

1. **Limited scope:** The theoretical analysis is restricted to quadratic functions, which may not capture the complexity of real-world neural networks.

2. **Performance in CV tasks:** OrtSepNorm shows suboptimal results in certain CV tasks (e.g., ResNet architectures), suggesting that the method’s advantages may be more task-dependent.

3. **Batch size:** The noise reduction in SepNorm relative to Lion is noted, yet it would be beneficial to have a more comprehensive analysis of batch size dependency across various architectures.

4. **Figures:** The paper lacks figures comparing the optimizers’ performance over time (epochs), which would provide insights into convergence speed and stability differences across tasks.

5. **No GitHub repository:** At this day, there is no open-source implementation provided.

While the paper provides hyperparameters used in experiments (in Appendix), it does not detail how these were selected (e.g., through grid search or prior literature) nor offer specific recommendations for practitioners aiming to apply these optimizers. Providing more details on the tuning process would enhance the reproducibility of the results. Moreover, additional experiments on smaller, well-known benchmarks like CIFAR-10/ CIFAR10-100 would provide clearer visualization of the optimizer's advantage and allow for direct performance comparisons, enhancing reproducibility and practical insight.

**Questions:**

1. Can the theoretical analysis extend beyond quadratic functions to provide deeper insights for non-convex loss landscapes?

2. How does the choice of parameter groups affect SepNorm’s performance across different neural network architectures?

3. Including the full algorithms for SepNorm and OrtSepNorm with detailed descriptions of hyperparameters, batch sizes, and optimization steps would improve clarity, enabling practitioners to reproduce the methods accurately and understand their setup more intuitively, maybe in the Appendix.

Typos :

“stabilise” -> “stabilize”

In the limitations, the phrase “to the training process” is repeated: “which in turn introduces instabilities to the training process. to the training process”

« Since the norm of the sign operation equals the number of nonzero elements…”  is repeated many times in the paper.

---

### Official Review · Reviewer_YKyW · 2024-11-01

**Soundness:** 2
**Presentation:** 2
**Contribution:** 2
**Rating:** 3
**Confidence:** 5

**Summary:**

This manuscript introduces an intermediary optimizer between signSGD and Norm-SGD, referred to as SepNorm. It includes some theoretical analyses under idealized assumptions and presents experimental results demonstrating that SepNorm achieves better performance on certain vision and language tasks compared to AdamW and Lion.

**Strengths:**

This manuscript  brings us attention to the effectiveness of Normalized SGD in training complex Transformers.

**Weaknesses:**

- The writing needs significant improvement. There are numerous grammatical errors throughout the manuscript that affect readability. The background on overparameterization (Lines 31-32 and 86-89) have tenuous connections to the proposed optimizer. In fact, overparameterization is typically absent in the training of large language models (LLMs); for example, Llama3-8b is trained with approximately 2000 tokens per parameter. Additionally, the references are not well cited or discussed. SepNorm builds on signSGD, yet the original signSGD paper and closely related works are not cited. At Line 100, the reference to classical Nesterov’s Accelerated Gradient (NAG) to underscore SGD is misleading, as NAG is not directly related to SGD. Furthermore, the numbering of Theorems in the main text is inconsistent with their proofs in the appendix.

- SepNorm appears to function more as an engineering trick. The only modification from Block Normalized Gradient (BNG) is the multiplication by the square root of the block size. In my previous experiments with training a ViT using both BNG and SepNorm on CIFAR-10, I found that, with finely tuned learning rates for both optimizers, there was no significant difference in performance.

- The theoretical analyses are based on unrealistic assumptions. For instance, Theorems 3.1, 4.1, and 4.2 require $ \langle \nabla F(w), w \rangle = 0 , \forall w $, which is impractical in real DNN training. Additionally, the analyses in Section 5 rely on an overly simplistic quadratic function, and Theorem 5.3 even requires \( A \) to be diagonal or to have identical eigenvalues. These unrealistic assumptions provide limited insights into the behavior of SepNorm in practical DNN training. It would be more beneficial to provide a theoretical convergence analysis for a new optimizer under milder assumptions.

- The experimental setup is also questionable. The baseline BNG is missing from all comparison experiments. Moreover, the experimental settings lack justification; for example, the learning rates for AdamW and Lion when training ViT in Table 6 are set to unusually low values of 6.25e-5 and 6.25e-6, which are significantly lower than those in the original Lion paper. The batch size for Lion is set to 256, much smaller than the optimal 4096. Additionally, the test accuracy in Table 1 is notably inferior to that reported in the Lion paper. It is also somewhat unusual that the model selected for the language task is T5, rather than the more popular GPT-like decode-only Transformers.

**Minor Issues**
- In Theorem 4.3, there is no definition of $d$  provided anywhere.
- The proof for Theorem 4.3 is omitted, and it may not be straightforward to obtain.

**Questions:**

N/A

---

### Official Review · Reviewer_X1FK · 2024-11-04

**Soundness:** 2
**Presentation:** 2
**Contribution:** 2
**Rating:** 3
**Confidence:** 5

**Summary:**

The paper shows the disadvantages of Lion and normalized gradient descent and proposes new methods called SepNorm and OrtSepNorm to address the disadvantages. Some theoretical insights are provided. Experimental results on computer vision (CV) and natural language processing (NLP) tasks seem to show the strengths of the proposed methods.

**Strengths:**

Analysis and improvement of the current adaptive methods such as Lion are important research topics.

**Weaknesses:**

1. The theory behind Lion (as described in Section 3) contains lots of hand-waving arguments that are not rigorous. For example, in line 160-line 161, the authors explained the possible benefit of Lion by the following statement: "Due to the vanishing gradient problem, some weights may receive small gradient components, especially those corresponding to first layers. " However, there is no evidence showing this is indeed the case or not. Similar hand-waving arguments also appear in line 197-line 208.

2. How is Theorem 3.1 related to the theory of Lion? I do not think there is a direct relationship. More explanations are needed.

3. In Section 4, the SepNorm seems to be a layer-wise (or group-wise) version of Normalized GD and Lion. However, it is unclear why Theorem 4.2 and Theorem 4.3 demonstrate the superior performance guarantees of the proposed methods. Also the proof can be trivially obtained by the property of scale-invariant networks.

4. Theorem 5.4 provides limited insights compared with normalized GD. How does the sharpness obtained by SepNorm imply the advantage over Lion and normalized GD? It is unclear to me. In addition, the proof of Theorem 5.4 seems to be a straightforward extension of [Arora et al. 2022].

5. In Section 6, there are missing details about how to select parameter groups for the proposed algorithms. In addition, the authors need to perform additional ablation studies to show the benefit of the proposed algorithms. For example, one baseline could be layer-wise optimizers such as LAMB (https://arxiv.org/pdf/1904.00962), which also uses group-wise (or layer-wise) learning rate. In addition, the experimental benefits of the proposed methods are marginal compared with existing optimizers such as AdamW.

**Questions:**

See weaknesses section.

---

### Official Review · Reviewer_dyv1 · 2024-11-05

**Soundness:** 3
**Presentation:** 3
**Contribution:** 3
**Rating:** 5
**Confidence:** 3

**Summary:**

This paper proposes a new optimization method that generalizes previous optimizers LION and normalized GD. The LION optimizer uses sign operations to perform the updates such that parameters have gradients of similar magnitudes. However,  the SIGN operation introduces more noises into the updates which may require a large batch size for good performance.  Normalized GD on the other hand do not have the issues from SIGN operation. However, some parameter groups may suffer from undertraining due to small gradients. To resolve these disadvantages, this paper proposes to normalize parameters by groups (note that the group size for LION is 1) while following the parameter updating rule of LION (named as SepNorm). It uses a quadratic model to show that the proposed methods obtain low loss values only when the sharpness (i.e., max eigenvalue of the hessian) is low for each parameter block. Finally, it performs experiments on vision transformers and language models to compare the method against LION and AdamW.

**Strengths:**

- This paper is clearly written. Most claims are supported with either theory or experiments. Figure 2  demonstrates the motivation of this work.

- The Grokking experiments are interesting, and it seems that OrtSepNorm (variations of SepNorm targeting scale-invariant networks) help to alleviate the issues (also see Thm 4.3).

**Weaknesses:**

It remains unclear to me the exact contributions of this work. If it is from the experimental side, it seems that the gain is marginal (e.g. Table 3). From the theoretical viewpoint, the analysis more or less follows that of SIGN and Normalized GD. It also lacks comparisons with previous theory works in section 5. In addition to this, the quadratic model perhaps is too simple to capture the underlying training dynamics.

**Questions:**

What are the technical challenges in doing the analysis?

---

### Meta-Review · Area_Chair_D48q · 2024-12-21

**Metareview:**

This paper studies a optimization method SepNorm which normalizes gradients across different parameter groups with the claim that it provides a more stable optimization dynamics compared to NormGD & Lion. The paper also conducts experiments on vision and NLP datasets to provide evidence for better quality. The authors also provide theoretical analysis for quadratic functions.

The reviews for the paper were mostly negative. The reviewers were mainly concerned regarding the (1) limited scope (2) weak empirical analysis (especially for the baselines) (3) informal arguments (4) unclear presentation. The authors did not respond to the reviewer's feedback. I recommend rejection in the current form.

**Additional Comments On Reviewer Discussion:**

The authors chose not to respond to the reviewer's feedback.

---

### Decision · Program_Chairs · 2025-01-22

Reject